# Antioxidant Benefits and Potential Mechanisms of Slightly Acidic Electrolyzed Water Germination in Sesame

**DOI:** 10.3390/foods12224104

**Published:** 2023-11-12

**Authors:** Yujie Li, Shaokang Liu, Jianxiong Hao, Huan Rao, Dandan Zhao, Xueqiang Liu

**Affiliations:** College of Food Science and Biology, Hebei University of Science and Technology, Shijiazhuang 050018, China; liyujie0604@163.com (Y.L.); 1345160280@163.com (S.L.); raohuan@hebust.edu.cn (H.R.); zdd6364@126.com (D.Z.); caulxq@163.com (X.L.)

**Keywords:** sesame, slightly acidic electrolytic water, antioxidant activity, antioxidant enzymes

## Abstract

Slightly acidic electrolytic water (SAEW) treatment for seed germination is a promising technique for sustainable agriculture. This study investigated the antioxidant activity of germinated sesame seeds treated with SAEW for the first time. Specifically, the impact and correlation of SAEW on the activities of total phenols, total flavonoids, and antioxidant oxidase in sesame seeds were examined. The results showed that SAEW with low ACC inhibited sesame germination, SAEW with high ACC promoted sesame germination, and sesame buds treated with SAEW with 30 mg/L and 50 mg/L ACC showed lower antioxidant activity and total phenolic and flavone content compared to tap water. In contrast, SAEW with 30 mg/L ACC had no significant effect on sesame growth but positively influenced the antioxidant activity of sesame seed germination by promoting phenolic compound synthesis through increased phenylalanine ammonia-lyase (PAL) activity and enhancing antioxidant activity by boosting PAL, polyphenol oxidase (PPO), and peroxidase (POD) activities. Generally, antioxidant ability was the most prominent in SAEW with 30 mg/L ACC, and positive correlations between antioxidation and total phenols and flavonoids content were found in sesame. These findings provide valuable insights into the mechanisms underlying the enhanced antioxidant capacity observed in germinated sesame seeds under SAEW stress.

## 1. Introduction

Free radicals are generated through natural body processes and by extrinsic factors such as radiation, pollution, and ultraviolet light [1]. Excessive formation of free radicals in the body can lead to the depletion of antioxidants, causing oxidative stress [2]. Oxidative stress is associated with numerous diseases, such as heart disease, diabetes, and certain types of cancer [3]. To mitigate the negative effects of oxidative stress, an adequate amount of antioxidants must be provided through a proper diet since they quench free radicals, thereby hindering oxidation. Epidemiological studies have shown that there is an inverse relationship between antioxidant intake and the occurrence of degenerative diseases [4]. Sesame contains a high amount of phenolic acids, flavonoids, vitamin e, and sesamol, with high antioxidant properties, which can clear free radicals, delay aging, and enhance immunity [5]. Therefore, improving the antioxidant activity of sesame and developing functional food hold promise and merit further investigation.

In response to the growing consumer demand for high nutritional quality, the food industry has witnessed the development of novel and reliable techniques. Among these techniques, germination has emerged as a safe, healthy, economic, and sustainable way to improve and enhance plant tissue structure, nutritional, and functional characteristics. An illustrative example is the germination of peanuts, which has been found to yield a plethora of bioactive compounds, including resveratrol, γ-aminobutyric acid (GABA), isoflavones, and polyphenol compounds. These substances offer various health benefits and contribute to the overall nutritional value of germinated peanuts [6]. Therefore, this processing strategy holds promise and merits further investigation. While the effects of germination on foxtail millet, proso millet, and common buckwheat have been extensively studied and documented, there is a noticeable lack of research focusing on the germination process of sesame seeds [7]. The limited availability of studies investigating the germination of sesame seeds underscores the need for further exploration in order to better understand the potential changes in antioxidant activity and nutritional composition that may occur during the germination of sesame seeds. In addition, seed germination under stress is a popular method to improve the antioxidant activity of plants. At present, there have been many reports on heavy metal [8], hypoxia [9], drought [10], and saline–alkali stress [11,12] seed germination, but the research on SAEW stress seed germination is obviously lacking.

Slightly acidic electrolyzed water (SAEW) was prepared by electrolyzing sodium chloride using a flow type electrolysis apparatus, and its application in sterilization was widely reported [13,14,15,16]. Research has demonstrated that the application of SAEW in the food industry exhibits a higher inactivation effect on microorganisms compared to several traditional treatments while causing minimal quality loss [17]. As a result, SAEW is considered a green, efficient, and safe disinfectant for food. In the process of enriching functional substances in sprouted grains, food safety issues are also taken into consideration. In the germination process, electrolytic water is being increasingly used as a medium, taking microbial action into consideration. Notably, research has shown that SAEW can inhibit the accumulation of abscisic acid content and reduce the accumulation of reactive oxygen species during watermelon germination [18]. Additionally, SAEW has been found to effectively reduce natural Salmonella contamination in alfalfa sprouts [19]. Moreover, it is found that SAEW germination could promote the contents of amino acids and phenolic compounds in brown rice and enhance its antioxidant activity [20]. Our previous results also demonstrated that SAEW could enhance the GABA accumulation of germinated buckwheat [21] and millet sprouts [22]. Considering the promising effects of SAEW on other grains and seeds, exploring its application in germinated sesame could uncover valuable insights into the enhancement of bioactive compound content and overall nutritional quality. Further research endeavors focusing on this aspect are warranted to bridge the knowledge gap and fully realize the potential benefits of SAEW germination in sesame.

The aim of this study was to determine the antioxidant activity of germinated sesame seeds treated with SAEW. Specifically, the impact and correlation of SAEW on the activities of total phenols, total flavonoids, and antioxidant oxidase in sesame seeds were examined. By exploring the effects and relationships between SAEW treatment and various biochemical markers, valuable insights can be gained regarding the optimization of sprouting conditions and the production of nutritionally enriched sesame products.

## 2. Materials and Methods

### 2.1. Sesame and Germination

The sesame used in this study belongs to oil sesame seeds with small particles, and was harvested in Shangqiu, Henan Province, China in 2022. To prepare sesame bud, the cleaned granule full sesame buds were soaked in the respective treatment water at solid-liquid ratio 1:5 and 30 °C for 1 h. Germination was carried out in the dark for 48 h (80% relative humidity) at 30 °C. To maintain humidity during germination, each treatment solution was sprayed separately on the sesame seeds every 12 h. Sesame bud samples were collected after 12, 24, 36, and 48 h and only soaked, but ungerminated sesame seeds were taken as 0 h sesame samples. At the same time, sesame seeds were germinated under the same germinating conditions with tap water instead of SAEW as control.

### 2.2. Treatment Solutions

SAEW was generated by the electrolysis of 8.6 mmol/L NaCl, 18 mmol/LCaCl_2_, and 125 μL/L HCl using a flow-type electrolysis apparatus (AQUACIDO NDX-250KMS, OSG Company Ltd., Osaka, Japan) equipped with a non-membrane electrolytic cell. The pH values of treatment solutions were measured with a pH meter (Orion Inc., Reston, VA, USA), and adjust the PH with HCl. The iodometric method was used to measure the available chlorine concentrations (ACC) in treatment solutions using a digital titrator (16900, Hach Company, Loveland, CO, USA). SAEW has a pH of 5.90 and an ACC of 10, 20, 30, 40, and 50. Tap water (TW) was used as control with a PH of 7.50.

### 2.3. Shoot Length, Germination Rate, and Fresh and Dry Weights of Sesame Bud

After sesame buds were harvested, 100 seeds were randomly selected to measure fresh weight, dried at 95 °C to constant weight to measure dry weight, and the germination number of sesame seeds was recorded. The germination rate (*GR*) of the sesame was calculated according to the following formulae:(1)$$GR\left(\%\right)=\frac{{NSG}_{n}}{TNS}\times 100,$$
where *GR* is the germination rate; *NSGn* is the number of seeds that germinated within *n* hours; *TNS* is the total number of seeds.

After 48 h of sesame germination, 30 seeds were randomly selected and the shoot length was measured with Vernier scale.

### 2.4. Extraction of Antioxidants

Extraction used our previous scheme [23]. Freeze-dried sesame buds (0.5 g) were fully ground with 15 mL methanol (12 mL methanol and 3 mL distilled water) (1:30 *w*/*v*) and shaken for 2 h in an electric shaker at 65 °C. Then, the filtrate was filtered with Whatman #4 filter paper, collected in a 25 mL volumetric bottle and fixed volume. The concentration of the stock solution of the sample was 20 mg/mL.

### 2.5. Antioxidant Assays

#### 2.5.1. DPPH Radical Scavenging Activity

DPPH assay was determined as described earlier [24] with some modification. Briefly, 1 mL sample extracts were mixed with 2 mL of freshly formulated 200 μM DPPH radical solution (100 mL 95% *v*/*v* methanol used to dissolve 7.9 mg DPPH). The absorbance at 517 nm was determined using the spectrophotometer after 30 min of incubation in the dark at room temperature. Then, the inhibition of free radical DPPH in percent was calculated according to the following formula:(2)$$Inhibition\left(\%\right)=\frac{A_{0}-A_{1}}{A_{0}}\times 100,$$
where *A*_0_ is the absorbance of DPPH and *A*_1_ is the absorbance of the sample.

#### 2.5.2. ABTS Radical Scavenging Activity

ABTS activity was measured as described earlier [24] with some modifications. The ABTS radical cation reagent was produced using a 7.4 mmol/L ABTS stock with a stock solution of 2.6 mmol/L of potassium persulphate (1:1, *v*/*v*). The solution was stored in the dark at room temperature for 12–16 h before prior use. The ABTS was then mitigated with methanol at 734 nm to obtain absorbance around 0.700~0.800. Later, 100 μL of diluted crude extracts or standards were merged into 1.9 mL of ABTS radical solution. The Spectrophotometer was used to monitor absorption at 734 nm after 60 min of incubation at room temperature in dark conditions. Then, the inhibition of free radical ABTS in percent was calculated according to the following formula:(3)$$Inhibition\left(\%\right)=\frac{A_{0}-A_{1}}{A_{0}}\times 100,$$
where *A*_0_ is the absorbance of ABTS and *A*_1_ is the absorbance of the sample.

#### 2.5.3. Hydroxyl Radical Scavenging Activity

Hydroxyl activity was measured as described earlier [24] with some modifications. Briefly, 0.5 mL of sample extract was treated with 0.5 mL of FeSO_4_, 0.5 mL of salicylic acid (9 mM 100 mL absolute ethyl alcohol used to dissolve 12.43 mg salicylic acid), 7 mL deionized water, and 0.5 mL of H_2_O_2_ (8.8 mM). The solution was incubated in a water bath at 37 °C for 15 min. The spectrophotometer was used to monitor absorption at 510 nm. Then, the inhibition of free radical hydroxy in percent was calculated according to the following formula:(4)$$Inhibition\left(\%\right)=\frac{A_{0}-A_{1}}{A_{0}}\times 100,$$
where *A*_0_ is the absorbance of the control and *A*_1_ is the absorbance of the sample.

#### 2.5.4. Ferric Reducing Antioxidant Power (FRAP)

FRAP was analyzed using the previously documented method [24] with some improvements. Briefly, 1 mL extracts were combined with a buffer of 2.5 mL (50 mM, pH 7) and 2.5 mL of K_3_[Fe(CN)_6_] (10 g/L), and the solution was incubated in a water bath at 50 °C for 20 min. After adding 2.5 mL trichloroacetic acid (100 g/L), the mixture was centrifuged at 4000× *g* for 10 min. Take 1 mL of supernatant and mix it with 3 mL of FeCl_3_ (10 g/L). The absorbance was measured at 700 nm. The degree of absorbance increase indicates the strength of reducing power.

### 2.6. Detection of Total Phenolic Content (TPC)

TPC was carried out using the previously mentioned protocol [25] with slight changes. Briefly, for a short duration of 8 min, 5 mL of sample extract or standard solution of gallic acid was treated with 2 mL of Folin-phenol reagent. The mixture was then alkalized with 3 mL of 10% Na_2_CO_3_. The absorbance was read at 735 nm after 120 min of incubation time at room temperature. Results were expressed as gallic acid (mg/g).

### 2.7. Detection of Total Flavonoid Content (TFC)

TFC was carried out using the previously mentioned protocol [25] with slight changes. Briefly, 1 mL of our sample extracts were added to 400 μL of NaNO_2_ (50 g/L). The reaction mixture was administered after 6 min of incubation with 400 μL AlCl_3_ (100 g/L). Further, 4 mL of NaOH (43 g/L) were added later after 6 min. The absorbance was read at 500 nm after 15 min of incubation time at room temperature. Results were expressed as rutin (mg/g).

### 2.8. Antioxidant Oxidase Analysis

Fresh sesame buds were fully ground with a refrigerated buffer containing EDTA and PVPP in an ice bath. The homogenates were centrifuged at 4 °C and the activities of superoxide dismutase (SOD), peroxidase (POD), catalase (CAT), ascorbate peroxidase (APX), phenylalanine ammonia-lyase (PAL), and polyphenol oxidase (PPO) enzymes were determined in the supernatant. Then, press the following formula to calculate the enzyme activity.

The antioxidant oxidase activity was determined using the previously documented method [26,27,28]. In this method, SOD reaction system included 50 mM phosphate buffer (pH 7.8), 100 μM EDTA, 130 mM L-methionine, 750 μM NBT, 20 μM riboflavin, and 0.6 mL samples. The absorbance was measured at 560 nm after 30 min reaction under light. The unit of SOD activity was defined as the amount of the enzyme required to inhibit 50% of the NBT photochemical reaction. Then, SOD activity was calculated according to the following formula:(5)SOD activityUg FW=Ack−Ae×VFW×Ack×Ve×0.5
where *FW* is the sample fresh weight; *A_ck_* is the absorbance of the control; *V_e_* is the amount of enzyme liquid used for determination; *A_e_* is the absorbance of the sample; and *V* is the total amount of enzyme liquid.

Similarly, APX reaction mixture included 50 mM phosphate buffer (pH7.8), 100 μM EDTA, 5 mM ascorbic acid, 0.1 mL of H_2_O_2_ (0.3%), and 0.2 mL of enzyme extract. CAT reaction mixture included 50 mM phosphate buffer (pH 7.0), 100 μM EDTA, 0.4 mL of H_2_O_2_ (0.3%), and 0.1 mL of enzyme extract. POD reaction mixture included 50 mM phosphate buffer (pH 7.8), 100 μM EDTA, 0.5 mL of guaiacol solution (0.3%), 0.2 mL of H_2_O_2_ (0.3%), and 0.2 mL of enzyme extract. PAL reaction mixture included 0.4 mL of enzyme extract, 2.6 mL of distilled water, and 1 mL PAL (20 mM). The mixture was then incubated for 1 h at 30 °C, and then a trifluoroacetic acid solution HCl (6 M, 0.2 mL) was added. PPO reaction mixture included 1 M catechol solution, 100 mM phosphate buffer (pH 6.8), and 20 μL of enzyme extract. The absorbance change in the solution was measured every 30 s at 290, 240, 470, 290, and 470 nm for 3 min, respectively. Enzyme activity unit was defined as the amount of enzyme required to change absorbance by 0.01 per minute. Then, the enzyme activity was calculated according to the following formula:(6)Enzyme activityUg FW=∆A×VFW×Ve×t×0.01
where *FW* is the sample fresh weight; *V_e_* is the amount of enzyme liquid used for determination; *t* is the response time; ∆A is the change in absorbance; and *V* is the total amount of enzyme liquid.

### 2.9. Statistical Analysis

Each treatment was performed in triplicate, and the resulting data were calculated as the mean and the standard deviation. All results were subjected to analysis of variance (ANOVA) and the means were compared with Duncan’s multiple range test and Pearson’s correlation test using the SPSS program (SPSS19.0 for Windows, SPSS Inc., Chicago, IL, USA). Statistical significance was set at a value of *p* < 0.05. The graphs were drawn using Origin 2022 drawing software.

## 3. Results and Discussion

### 3.1. Effect of SAEW on Shoot Length, Germination Rate, Fresh and Dry Weight, and Germination Rate of Sesame

The shoot length, germination rate, and fresh and dry weights of sesame seeds treated with SAEW at different ACC were shown in Figure 1. As can be seen from the graph, the length of sesame sprouts increased with the increase in ACC, but, when ACC was lower, sesame sprouts were shorter, germination rate was lower, fresh weight was lighter, and the growth of sesame was inhibited; when ACC was high, the growth of sesame was promoted, and the dry matter had no significant difference among different solutions. This is mainly because the difference in ACC affects the activity of catalase and peroxidase and changes the stress resistance of sesame seeds; therefore, there are differences in sesame bud among different treatments. In addition, SAEW prolonged the dormant period of seeds and did not cause the decomposition and synthesis of a large number of substances within 48 h of sesame germination, resulting in no obvious change in dry weight. Conversely, Hao et al. found that SAEW with higher ACC inhibited the growth of buckwheat and broccoli flower buds [23,29]. The difference in SAEW’s effect on growth is mainly due to the difference in species. In addition, buds adapt to osmotic stress caused by ACC, water swelling, increased intracellular enzyme activity, material decomposition, and new synthesis for growth and germination. Further studies are needed to elucidate the role of SAEW of different ACC in sesame bud cells.

### 3.2. Effect of SAEW on Antioxidant Activities of Germinated Sesame Seeds

In living organisms, the generation of free radicals through oxidation and their subsequent elimination through antioxidants maintain a delicate balance under normal conditions. However, an imbalance in oxidation–reduction can lead to an excessive accumulation of free radicals, which adversely affects cellular activities, alters protein structure, deactivates enzymes, disrupts hormone function, and compromises the immune system. This cumulative damage has detrimental effects on the human body, contributing to various diseases, such as cancer, arteriosclerosis, cardiovascular disorders, and cerebrovascular diseases. In order to counteract these detrimental effects, plants enhance both enzymatic antioxidant activity and non-enzymatic antioxidant components [30].

Research has demonstrated that the germination of millet leads to a notable rise in the levels of phenols, flavonoids, and GABA. This increase subsequently contributes to enhanced antioxidant activity observed in vitro [31]. Li et al., studied iron fortification in germinated brown rice and its association with total phenols and flavonoids. They observed significant increases in both, positively correlated with DPPH and ABTS scavenging activities. However, no significant correlation was found with iron reducing power (FRAP) values [32]. Sharma et al., examined eight types of germinated barley, noting increased antioxidant activity positively correlated with total phenol content [33]. Juan et al. observed increased CAT activity during tobacco seed development, related to oxidative stress [34]. Ma et al., explored the correlation between PAL, C4H, 4CL, antioxidant enzymes (POD, CAT, SOD, APX, GR, GST), and antioxidant activities during barley germination under NaCl stress [35]. For sesame seeds, further studies are needed to determine whether the antioxidant activity of SAEW-germinated sesame seeds can be enhanced by increasing the content of active substances or by boosting the activity of antioxidant enzymes. Additionally, there is a need for further exploration to identify which antioxidant enzymes are affected by SAEW.

In this study, we investigated the impact of different concentrations of ACC in SAEW on the germination of sesame seeds. We aimed to explore the free radical scavenging ability of SAEW and its relationship with enzymatic and non-enzymatic antioxidants. As shown in Figure 2, the ability of SAEW to scavenge DPPH free radicals was continuously enhanced during germination. Initially, SAEW with different ACC exhibited significant differences in their scavenging abilities, but, after 48 h, the DPPH scavenging abilities became similar across all concentrations. However, it is worth noting that, within the first 48 h of germination, SAEW with an ACC of 30 mg/L demonstrated a greater extent of DPPH scavenging compared to TW and other ACC. Our assessment of the ability of SAEW to scavenge hydroxyl radicals showed a similar trend to that of DPPH. The scavenging activity of ABTS was found to be higher in the germinated sesame samples treated with SAEW20 (65.78 ± 0.19%), SAEW30 (77.52 ± 0.93%), and SAEW40 (75.55 ± 0.11%) compared to those treated with TW (57.66 ± 0.77%) after 48 h. Raw sesame samples exhibited the lowest ABTS activity. Similar trends were observed in our study assessing the Ferric Reducing Antioxidant Power (FRAP). The FRAP values were also higher in the germinated samples treated with SAEW30 (0.493 ± 0.006) and SAEW40 (0.461 ± 0.003) compared to those treated with TW (0.428 ± 0.005) after 48 h.

SAEW with different ACC exhibited varying free radical scavenging abilities. Specifically, SAEW50 initially showed a low scavenging ability in the early stage of germination, but this ability was significantly enhanced after 48 h of germination. This suggests that, under the influence of SAEW50, the metabolism of endogenous germination inhibitors in the seeds slowed down, leading to an increase in seed dormancy period. As the germination time progressed, the seeds continued to absorb water, which enhanced their respiratory activity. This transition from dormant static state to active physiological activity resulted in the activation of numerous enzymes and increased the ability to scavenge free radicals within the seeds.

Proper concentration of available chlorine in SAEW can enhance the scavenging ability of free radicals. However, different seed types and varieties may respond differently to ACC. In this study, the scavenging ability of SAEW50 was initially low in the early stage of germination but significantly increased after 48 h of germination. As the germination time progressed, the seeds continued to absorb water, leading to enhanced respiration and a transition from a static dormant state to a dynamic state of frequent physiological activities. This activation of enzymes resulted in an improved ability to remove free radicals within the seeds. Under SAEW30 stress, the seed dormancy time was shortened, cell metabolic activity became stronger, and the seed’s REDOX reaction increased, leading to a higher ability to scavenge free radicals. In the next step, correlation analysis will be conducted to determine the relationship between the scavenging capacity of free radicals and the TPC and TFC, as well as antioxidant enzyme activity.

### 3.3. Effect of Germination on the TPC and TFC

Phenolic substances found in plants possess significant antioxidant activities, primarily by chelating active metal ions and inhibiting lipid peroxidation. These compounds play a crucial role in the germination process of plants, serving both structural growth purposes and providing protection against various abiotic and biotic stresses. The synthesis of phenolic compounds during germination is essential for the overall development and defense mechanisms of the plant [36].

Figure 3 showed the changes in TPC and TFC of germinated sesame seeds treated with SAEW (ACC = 20, 30, 40) during the germination process. The total phenolic content showed a gradual increase from 0.23 ± 0.03 to 1.89 ± 0.01 mg/g as germination progressed. Specifically, under the stress of SAEW20, 30, and 40, the increase in total phenolic content was 0.247, 0.765, and 0.582 mg/g higher, respectively, compared to TW stress at 48 h of germination.

In contrast, the TFC initially decreased and then increased during germination. Under SAEW30 and 40 stress, the increase in TFC was 0.754 and 0.730 mg/g higher, respectively, compared to TW stress at 48 h of germination. Therefore, SAEW, as a stress condition, may be involved in regulating the activity of phenylalanine ammonia-lyase (PAL), a key enzyme in the metabolic pathway of phenylpropane compounds. This regulation then affects the levels of total phenols and total flavonoids. Further exploration is needed to understand the underlying mechanisms [37].

### 3.4. Effect of Germination on Antioxidant Enzymes Activity

According to the above results of antioxidant activity and total phenolic flavonoid content of sesame, it can be seen that SAEW10 and SAEW50 had low antioxidant capacity and significantly inhibit the synthesis of phenolic substances. Therefore, to investigate the reasons behind the increase in antioxidant activity, we examined the changes in enzymatic activity of SOD (superoxide dismutase), POD (peroxidase), CAT (catalase), APX (ascorbate peroxidase), PAL (phenylalanine ammonia-lyase), and PPO (polyphenol oxidase) in sesame seeds treated with SAEW20, 30, 40. As depicted in Figure 4, the trend of antioxidant oxidase activity in germinated sesame seeds treated with SAEW was similar to that of the control group, showing an overall increase with germination time. However, there were a few differences observed, such as a slight decrease in SOD activity and a peak in CAT activity at 36 h. In general, both SAEW20 and SAEW30 treatments exhibited better performance compared to the TW control group regarding antioxidant oxidase activity in sesame seeds.

The presence of reactive oxygen species (ROS) such as HClO, ClO_2_^−^, ·OH, H_2_O_2_, and O_2_^−^ O in SAEW, along with the increase in ACC (presumably referring to the active chlorine concentration), can contribute to the divergent effects observed in comparison to TW treatments. These ROS molecules are known to have different oxidative properties and can impact the activity of antioxidant enzymes in sesame sprouts. This can lead to variations in the response of the antioxidant defense system, resulting in different outcomes between SAEW and TW treatments. SAEW20 and SAEW30 treatments led to increased PPO activity, promoting enzymatic browning, while seedling APX activity decreased. Conversely, SOD, POD, and CAT activities were higher compared to TW treatment, indicating a response to mitigate oxidative stress induced by SAEW-treated seeds. Additionally, during germination, hypochlorous acid, the primary component of SAEW, may undergo conversion into chloride [38]. Previous studies showed that chloride might stimulate the oxidative response of plants, and research has shown that salt stress alters the amount and the activities of the enzymes involved in ROS scavenging [39,40]. In summary, SAEW treatment has been shown to enhance antioxidant activity and stimulate the germination and growth of sesame buds. These effects are associated with a decrease in APX activity and an increase in SOD, POD, CAT, and PPO activity in the treated sesame buds. The presence of ACC and ROS in SAEW is believed to contribute to the observed antioxidant activity and growth-promoting effects in sesame buds.

### 3.5. Correlation between the Antioxidant Activities, the Total Phenolic, Flavonoid, and Activity of Antioxidant Enzyme

According to the above sesame growth characteristics, antioxidant activity, total phenolic, and flavonoid content and antioxidant enzyme activity during sesame germination, SAEW30 was the most suitable for sesame germination. Therefore, it is meaningful to compare and analyze the correlation between SAEW30 and tap water to explore the mechanism of SAEW30 promoting antioxidant activity in sesame germination. Pearson’s correlation coefficient was used to link the antioxidant activities measured by the different methods with each other, as well as with the total phenolic, flavonoid, and activity of antioxidant enzyme of the extracts. Figure 5 displays the correlations between total phenolic, flavonoid, and activity of antioxidant enzyme and antioxidant activities in germination of sesame under SAEW30 and the TW. In this study, statistical analysis of the data gathered above showed that SAEW stressed sesame germination, total phenolic, flavonoids content, and PPO, and PAL enzyme activities and ABTS free radical scavenging ability increased correlation. The activity of POD, APX, PAL, and PPO enzymes increased correlation with the iron reduction ability. The regression analysis showed that the change in total phenol content was positively correlated with PAL activity. The increase in total phenol content during 48 h of germination may be caused by the stimulation of PAL activity after SAEW stress, which can stimulate the synthesis of phenolics in the phenylpropane metabolic pathway to repair oxidative stress damage [41]. Therefore, the PAL activity and total phenol content induced by SAEW-germinated sesame seeds were higher, which enhanced the ability of cells to resist external oxidative damage.

## 4. Conclusions

Five slightly acidic electrolyzed waters with different available chlorine concentrations (10, 20, 30, 40, and 50 mg/L) were tested to evaluate their responses to growth characteristics, antioxidant activity, total phenol, flavonoid, and antioxidant enzyme during sesame germination. The results showed that SAEW with low ACC inhibited sesame germination, SAEW with high ACC promoted sesame germination, and sesame buds treated with SAEW10 and SAEW50 showed lower antioxidant activity and total phenolic and flavone content compared to the tap water. On the whole, SAEW with an ACC of 30 mg/L exhibited the highest antioxidant activity and antioxidant enzyme activity in germinated sesame sprouts. Additionally, sesame seeds germinated with SAEW contained abundant polyphenols and flavonoids. Enzyme production and core changes occur during the germination process, which contribute to the elevation of intrinsic phenolic compounds and antioxidant activity in sprouted sesame seeds. In our study, SAEW stress positively influences the antioxidant activity of sesame seed germination by promoting phenolic compound synthesis through increased PAL activity and enhancing iron reduction through the activities of enzymes like POD, APX, PAL, and PPO. These findings provide valuable insights into the mechanisms underlying the enhanced antioxidant capacity observed in germinated sesame seeds under SAEW stress. These findings highlight the potential of SAEW as a beneficial method for enhancing the antioxidant properties and bioactive compound content of germinated sesame seeds. Further research could delve into optimizing the ACC level of SAEW and investigating the underlying mechanisms behind the observed effects on cell breakdown and antioxidant activity during sesame germination. By exploring the effects and relationships between SAEW treatment and various bio-chemical markers, valuable insights can be gained regarding the optimization of sprouting conditions and the production of nutritionally enriched sesame products.

## Figures and Tables

**Figure 1 foods-12-04104-f001:**
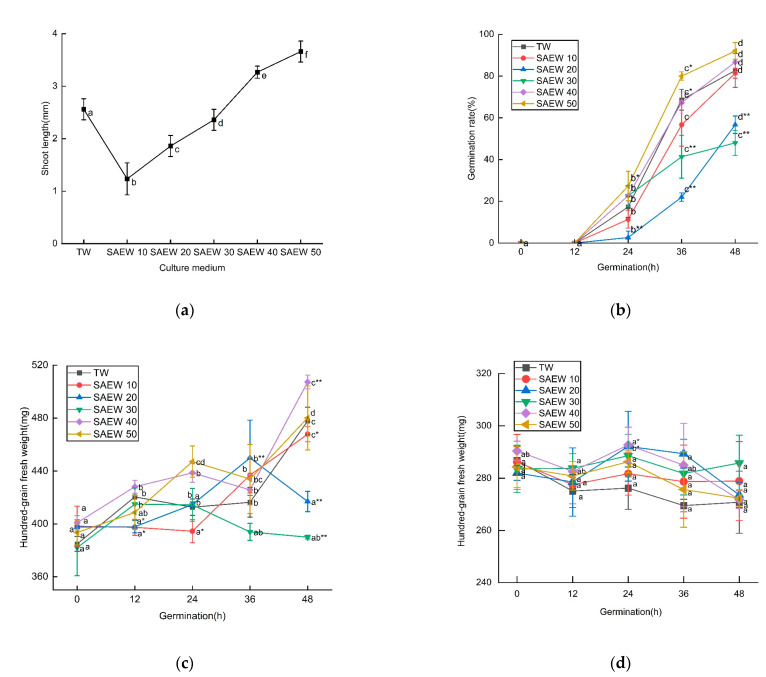
The changes in (**a**) shoot length at 48 h, (**b**) germination rate, (**c**) fresh weight, and (**d**) dry weight of 100 grains during germination of sesame treated by slightly acidic electrolyzed water (SAEW) during germination. The pH and available chlorine concentration (ACC) of SAEW used in the experiment were 5.9 ± 0.1 and 10, 20, 30, 40, 50 mg/L, respectively; the tap water (TW) as control was the local drinking water. Samples were taken at 0, 12, 24, 36, 48 h of germination. Each value is expressed as the mean ± standard deviation of three replicates. Different superscripts (a–f) show significant difference in duration of germination (*p* < 0.05). SAEW is significantly different from the control and is indicated by * (*p* < 0.05) and ** (*p* < 0.01).

**Figure 2 foods-12-04104-f002:**
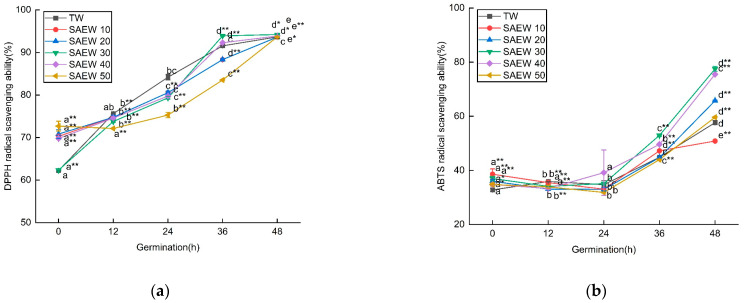
The changes in (**a**) DPPH, (**b**) ABTS, (**c**) hydroxyl radical scavenging ability, and (**d**) Fe^3+^ reducing ability of sesame treated by slightly acidic electrolyzed water (SAEW) during germination. The pH and available chlorine concentration (ACC) of SAEW used in the experiment were 5.9 ± 0.1 and 10, 20, 30, 40, 50 mg/L, respectively; the tap water (TW) as control was the local drinking water. Samples were taken at 0, 12, 24, 36, 48 h of germination. Each value is expressed as the mean ± standard deviation of three replicates. Different superscripts (a–e) show significant difference in duration of germination (*p* < 0.05). SAEW is significantly different from the control and is indicated by * (*p* < 0.05) and ** (*p* < 0.01).

**Figure 3 foods-12-04104-f003:**
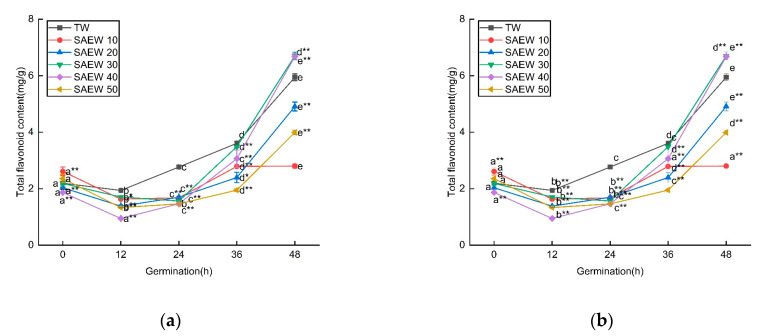
The changes in (**a**) total phenolic contents (TPC) and (**b**) total flavonoid contents (TFC) of sesame treated by slightly acidic electrolyzed water (SAEW) during germination. The pH and available chlorine concentration (ACC) of SAEW used in the experiment were 5.9 ± 0.1 and 20, 30, 40 mg/L, respectively; the tap water (TW) as control was the local drinking water. Samples were taken at 0, 12, 24, 36, 48 h of germination. Each value is expressed as the mean ± standard deviation of three replicates. Different superscripts (a–e) show significant difference in duration of germination (*p* < 0.05). SAEW is significantly different from the control and is indicated by * (*p* < 0.05) and ** (*p* < 0.01).

**Figure 4 foods-12-04104-f004:**
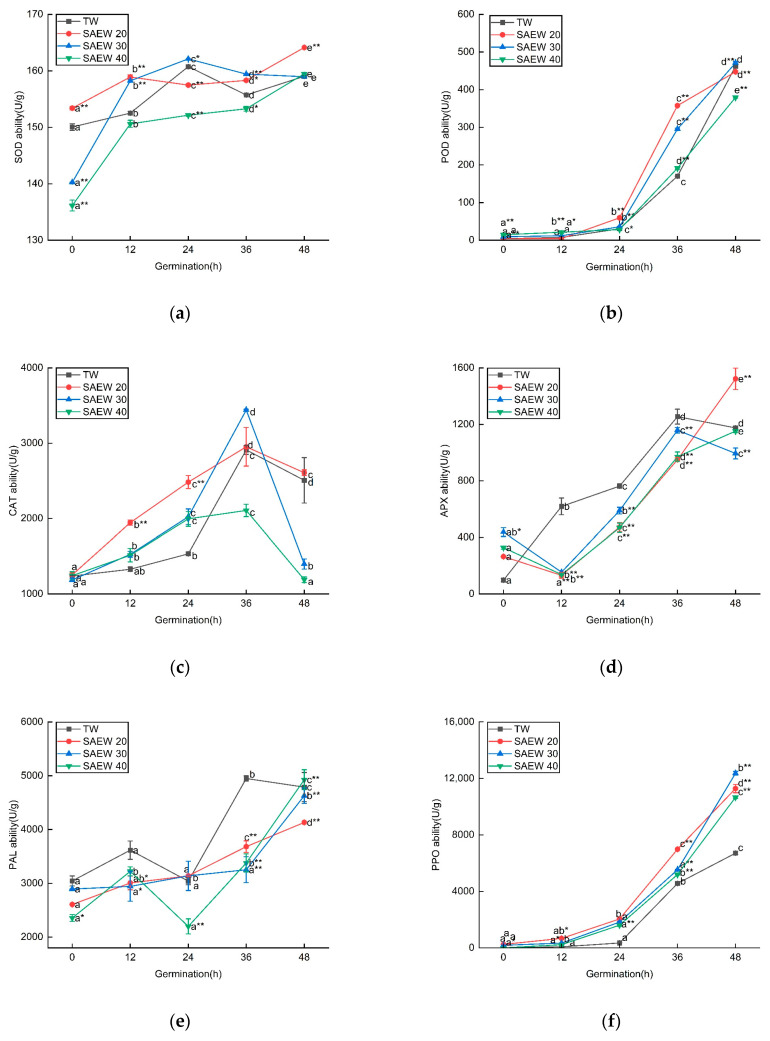
The changes in (**a**) SOD, (**b**) POD, (**c**) CAT (**d**) APX, (**e**) PAL, (**f**) PPO activities of different antioxidant enzymes of sesame treated by slightly acidic electrolyzed water (SAEW) during germination. The pH and available chlorine concentration (ACC) of SAEW used in the experiment were 5.9 ± 0.1 and 20, 30, 40 mg/L, respectively; the tap water (TW) as control was the local drinking water. Samples were taken at 0, 12, 24, 36, 48 h of germination. Each value is expressed as the mean ± standard deviation of three replicates. Different superscripts (a–e) show significant difference in duration of germination (*p* < 0.05). SAEW is significantly different from the control and is indicated by * (*p* < 0.05) and ** (*p* < 0.01).

**Figure 5 foods-12-04104-f005:**
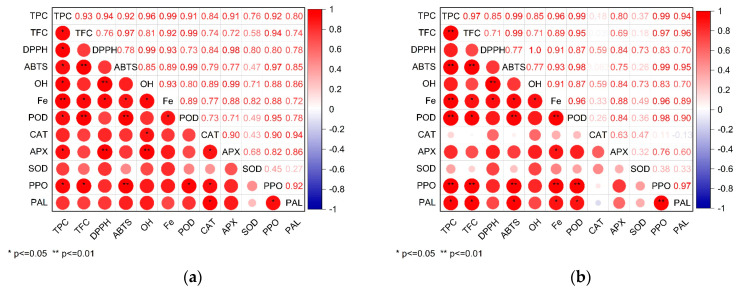
The Pearson correlation coefficient changes in antioxidant activities, the total phenolic, flavonoid, and activity of antioxidant enzyme of sesame seeds during germination under (**a**) the tap water (TW) and (**b**) slightly acidic electrolyzed water (SAEW). The pH and available chlorine concentration (ACC) of SAEW used in the experiment were 5.9 ± 0.1 and 30 mg/L.

## Data Availability

The data are available from the corresponding author.

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
