# Peer review of "Antioxidant Benefits and Potential Mechanisms of Slightly Acidic Electrolyzed Water Germination in Sesame"

_foods, 2023, doi:10.3390/foods12224104_

Round 1
Reviewer 1 Report
Comments and Suggestions for Authors
2.1. Overview and general recommendation
In this work, the authors test different concentrations of slightly acidic electrolyte water (SAEW) to “biofortify” sesame seeds. The results are interesting, but there are many doubts about how the work was carried out. Mainly with regard to the use of controls with different pH, and the statistical method used to evaluate the effect of treatments and germination time.
Furthermore, the difficulty in visualizing the data in the format of the graphs presented does not allow us to reach the same conclusion as the authors (Best treatment - SAEW30)
2.2 Major considerations
1. Lines 88-95. Treatment solutions (2.2): From what I understand, the flow-type electrolysis device used generates SAEW with different ACC and similar pH (pH correction was not mentioned) from the same solutions. Even so, pH analysis is performed on the produced SAEW and the control solution (Tap water). If there is this control with pH, it is understood that the authors believe that pH may be a confounding factor for the results of the work. That said, why was a control treatment used with a very different pH from the treatments tested?
2. Line 107. Is methanol produced by mixing ethanol and water? The article cited by the authors (16) uses 80% methanol.
3. Lines 195-199. What statistical analyses were used? The authors mention three but only talk about Pearson's correction. However, they present data that appears to be from a one-way ANOVA. I suggest using two-way ANOVA, with the factors of time and treatment.
4. Figure 1. There is no citation of results in which there was no difference between the tested treatments and the control ((b) germination rate, (c) fresh mass and (d) dry mass of 100 grains during sesame germination. You do not think this is necessary discuss this data?
5. It is not possible to view standard deviations on all Figures. I suggest using a bar chart.
6. Figure 4. Why are results not shown for all treatments tested? I didn't see an explanation about this in the text.
7. Why was the Pearson correlation performed only with the SAEW30 treatment?
2.3. Minor considerations
8. Lines 74-76. This excerpt can be used in the discussion of the manuscript.
Author Response
请参阅附件。

Reviewer 2 Report
Comments and Suggestions for Authors
Sesame is a valuable crop with many medicinal potentials and applications. So, the improving of total phenolic and flavonoid contents is something great but the authors should expand their research to investigate the content of the produced seeds’ phytochemicals, oil and proteins.
Reviewer 3 Report
Comments and Suggestions for Authors
The authors investigated the antioxidant activity of germinated sesame seeds treated with SAEW for the first time. Specifically, the impact and correlation of SAEW on the activities of total phenols, total flavonoids, and antioxidant oxidase in sesame seeds were examined. The results showed that SAEW with 13 mg/L available chlorine concentrations (ACC) promoted sesame germination and positively influenced the antioxidant activity of sesame seed germination by promoting phenolic compound synthesis through increased phenylalanine ammonia-lyase (PAL) activity and enhancing antioxidant activity by boosting PAL, polyphenol oxidase (PPO) and peroxidase (POD) activities. This manuscript should be carefully revised before being processed further in the journal. Some suggestions have been shown below.
Moderate English editing and spell check is required. A few typographical errors also need to be addressed.
Comments:
Abstract: Only general statements are written; they should present the results and findings. Rewrite the abstract.
Introduction: The introduction seems more like a review and no connection between subsections. It should be precise and highlight the importance of the present study.
Keywords: solar radiations and co‐existing species, germination and growth profiles; they should be replaced.
Materials and Methods:
What standard procedure was followed for seed germination?
Adjust the spacing error throughout the manuscript.
What was the EC of TW used as a control?
Did the authors use TW for making different treatments? Didn’t TW interfere with said concentrations?
L238: What is the reference number of Li et al.??
Conclusions: It should be very precise. Rewrite it.
L391: There should be no references in the conclusion section.
Comments on the Quality of English Language
Moderate English editing and spell check is required.
Reviewer 4 Report
Comments and Suggestions for Authors
Yujie Li et al. present mechanism of SAEW implication in germination in sesame in their manuscript. It was first SAEW using in improve on production and nutrionallity sesame seeds. I am only two inquries to metodical section. The results was presentd clearly. And the discussion section corresponds with results. Moreover, the authors citations novelty publications publish after 2002 year.
My questions are:
1. In material and methods sections authors indicate that control samples are ungerminated sesame seeds. Please explain why the sesame seeds treatment by clear water wasn't used as control?
2. In assays 2.5.1 and 2.5.2 authors indicate that A0 is the absorbance of DPPH and ABTS, respectively. IN my opinion A0 should be a absorbation of control sample (DPPH/ABTS + solvent). Please explain.
Round 2
Reviewer 1 Report
Comments and Suggestions for Authors
All questions were answered by the authors